# Advanced Spectroscopic Studies of the AIE-Enhanced ESIPT Effect in a Selected 1,3,4-Thiadiazole Derivative in Liposomal Systems with DPPC

**DOI:** 10.3390/ijms262110643

**Published:** 2025-10-31

**Authors:** Alicja Skrzypek, Iwona Budziak-Wieczorek, Lidia Ślusarczyk, Andrzej Górecki, Daniel Kamiński, Anita Kwaśniewska, Sylwia Okoń, Igor Różyło, Arkadiusz Matwijczuk

**Affiliations:** 1Department of Chemistry, Faculty of Food Sciences and Biotechnology, University of Life Sciences in Lublin, Akademicka 15, 20-950 Lublin, Poland; alicja.skrzypek@up.lublin.pl; 2Department of Biophysics, Faculty of Environmental Biology, University of Life Sciences in Lublin, Akademicka 13, 20-950 Lublin, Poland; lidia.slusarczyk@up.lublin.pl; 3Department of Physical Biochemistry, Faculty of Biochemistry, Biophysics and Biotechnology, Jagiellonian University, Gronostajowa 7, 30-387 Kraków, Poland; andrzej.gorecki@uj.edu.pl; 4Department of Chemistry, Maria Curie-Sklodowska University, Akademicka 19, 20-033 Lublin, Poland; daniel.kaminski@mail.umcs.pl; 5Department of Applied Physics, Faculty of Mechanical Engineering, Lublin University of Technology, Nadbystrzycka 38, 20-618 Lublin, Poland; a.kwasniewska@pollub.pl; 6Institute of Plant Genetics, Breeding and Biotechnology, University of Life Sciences in Lublin, 20-950 Lublin, Poland; sylwia.okon@up.lublin.pl; 7Faculty of Medicine, Medical University of Lodz, Al. Kościuszki 4, 90-419 Łódź, Poland; rozyloigor@gmail.com

**Keywords:** 1,3,4-thiadiazole (NTBD), ESIPT + AIE, dual fluorescence, molecular spectroscopy, liposomal system

## Abstract

Liposomal systems are advanced carriers of active substances which, thanks to their ability to encapsulate these substances, significantly improve their pharmacokinetics, bioavailability, and selectivity. This article presents the results of spectroscopic studies for a selected compound from the 1,3,4-thiadiazole group, namely 4-[5-(naphthalen-1-ylmethyl)-1,3,4-thiadiazol-2-yl]benzene-1,3-diol (NTBD, see below in the text), in selected liposomal systems formed from the phospholipid *1,2-dipalmitoyl-sn-glycero-3-phosphocholine* (DPPC). Detailed spectroscopic analyses were carried out using electronic absorption and fluorescence spectroscopy; resonance light scattering (RLS) spectra measurements; dynamic light scattering (DLS); as well as time-resolved methods—fluorescence lifetime measurements using the TCSPC technique. Subsequently, based on the interpretation of spectra obtained by FTIR infrared spectroscopy, the preliminary molecular organization of the above-mentioned compounds within lipid multilayers was determined. It was found that NTBD preferentially occupies the region of polar lipid headgroups in the lipid multilayer, although it also noticeably interacts with the hydrocarbon chains of the lipids. Furthermore, X-ray diffraction (XRD) techniques were used to study the effect of NTBD on the molecular organization of DPPC lipid multilayers. Monomeric structures and aggregated forms of the above-mentioned 1,3,4-thiadiazole analogue were characterized using X-ray crystallography. Interesting dual fluorescence effects observed in steady-state fluorescence measurements were linked to the excited-state intramolecular proton transfer (ESIPT) effect (based on our earlier studies), which, in the obtained biophysical systems—liposomal systems with strong hydrophobicity—is greatly enhanced by aggregation-induced emission (AIE) effects. In summary, the research presented in this study, concerning the novel 1,3,4-thiadiazole derivative NTBD, is highly relevant to drug delivery systems, such as various model liposomal systems, as it demonstrates that depending on the concentration of the selected fluorophore, different forms may be present, allowing for appropriate modulation of its biological activity.

## 1. Introduction

Liposomal systems are versatile and highly valuable platforms in biology, medicine, agriculture, and the food industry. In biological and biomedical research, liposomes serve as model membranes and drug delivery systems due to their ability to encapsulate diverse molecules, improve solubility, and enhance bioavailability, while minimizing toxicity [1,2,3,4,5]. In agriculture, liposomes have emerged as eco-friendly carriers of pesticides and fertilizers, enabling controlled release and reducing environmental impact. Similarly, in the food industry, liposomal formulations improve the stability and bioavailability of nutrients, extend product shelf life, and allow the use of natural components without synthetic additives [6,7,8]. Knowledge of the molecular form of a given drug in such a biophysical system is crucial and may influence the potential applications of the compound.

A key advantage of liposomal systems is their tunable microenvironment, which can modulate the behavior of encapsulated molecules. This feature is particularly relevant for fluorescent compounds exhibiting photophysical effects such as aggregation-induced emission (AIE) and excited-state intramolecular proton transfer (ESIPT). Molecules showing AIE are non-emissive in dilute solutions but display strong fluorescence upon aggregation, making them ideal for sensing and imaging applications in complex biological or lipid-based environments [9,10,11]. Aggregation-induced emission (AIE) is a photophysical phenomenon in which certain molecules, non-emissive in dilute solution, exhibit strong fluorescence upon aggregation. This effect has been widely exploited in designing functional materials for sensing, imaging, and bioanalytical applications [12,13].

According to literature data, compounds that represent a fairly promising source of derivatives exhibiting a range of interesting biological as well as photophysical properties are those from the 1,3,4-thiadiazole group [14]. This group demonstrates many promising antimicrobial, anticancer, and neuroprotective properties [15,16], with many of its derivatives already in advanced stages of clinical trials or even in medical use [17]. From a technical standpoint, various 1,3,4-thiadiazole derivatives also find many other noteworthy applications, including in the design of novel lasers [18], as oxidation inhibitors [19], or as ligands effectively chelating various d-block metal ions [20].

From a spectroscopic perspective, a particularly interesting effect exhibited by this group of compounds is the excited-state intramolecular proton transfer (ESIPT) process [21], which in fluorescence emission spectra most often manifests as the phenomenon of dual fluorescence resulting from a single excitation [22,23]. The dual fluorescence effect itself is described in the literature as arising from the coexistence of multiple emissive states in different electronic and/or molecular structures, associated with the aforementioned ESPT [24] or ESIPT processes. In addition, molecules exhibiting twisted intramolecular charge transfer (TICT) processes [25]—often accompanied by violations of Kasha’s rule [26]—are also very common, as are aggregation effects related to the formation of excimer systems [27] or the increasingly widespread aggregation-induced emission (AIE) and its enhancement (AIEE), particularly associated with the restriction of intramolecular rotation (RIR) mechanism [28].

Returning to the selected, specific group of 1,3,4-thiadiazoles with 2,4-dihydroxyphenyl substituents, dual fluorescence has most often been explained—by our group as well as others—by referring to effects related to the ESPT process, specifically excited-state intramolecular proton transfer (ESIPT) [22]. Moreover, in our detailed spectroscopic and theoretical studies, we have observed that the ESIPT process in these systems is particularly strongly enhanced by aggregation effects associated with AIE-type fluorescence [23,29]. We believe that analysis of the mechanisms underlying these spectroscopic phenomena can provide valuable insights into the electronic and conformational behavior of the studied 1,3,4-thiadiazole derivatives, which in turn may help explain their biological activity. The ESIPT process has been the subject of intensive research for several decades, contributing to numerous practical applications. Molecules exhibiting this effect are used as, for example: laser dyes [30], solar cells [31], light-emitting diodes (LEDs) [32], sensors [33] and polymer stabilizers protecting against UV light [34]. An increasingly interesting application of ESIPT molecules involves their use as molecular probes in advanced biological systems related to proteins or DNA [35]. It is also worth mentioning the proton transfer-triggered proton transfer (PTTPT) process, in which an initial ESIPT initiates a secondary ESIPT, opening new functional possibilities for molecules exhibiting the ESPT process [36].

In the present work, we explored the molecular mechanisms of fluorescence effects occurring in a 1,3,4-thiadiazole derivative with pharmacological properties previously confirmed in earlier studies [22,23,37], namely 4-[5-(naphthalen-1-ylmethyl)-1,3,4-thiadiazol-2-yl]benzene-1,3-diol (NTBD) (Figure 1), in model biological systems such as liposomal assemblies. The NTBD structure favors intramolecular hydrogen bonding between the proton-donating group (-OH in the resorcinol ring) and the proton-accepting group (N atom in the 1,3,4-thiadiazole ring), which is a structural requirement for tautomerism associated with the ESIPT process. Building on our previous studies of this derivative and considering the nature of the selected environment, characterized by high hydrophobicity, this work emphasizes the importance of aggregation effects related to AIE-type fluorescence and their influence on inducing dual fluorescence depending on the type of environment.

This article therefore presents a more comprehensive study of the above-mentioned fluorescence effects in liposome-type biophysical systems, employing various spectroscopic techniques, including steady-state absorption spectroscopy, fluorescence spectroscopy (with resonance light scattering, RLS), and time-resolved fluorescence lifetime measurements using the time-correlated single photon counting (TCSPC) technique. Additionally, FTIR infrared spectroscopy studies reveal the main locations of the selected 1,3,4-thiadiazole within lipid membranes formed from DPPC. FTIR results clearly indicate that NTBD molecules interact with both the hydrophobic and hydrophilic parts of the lipid membrane, although interaction with the polar region of the lipid bilayer appears to be significantly more favored for this derivative.

The analysis primarily demonstrated the interdependence between excited-state proton transfer and molecular aggregation processes associated with the AIE phenomenon, which very effectively suppress non-radiative deactivation pathways that could limit the dual fluorescence effects of NTBD. It was shown that the liposomal environment has a significant influence on the fluorescence effects of this thiadiazole derivative. This suggests that even slight changes in the microenvironment can lead to substantial modifications of the photophysical properties of these compounds, and the resulting differences in their electronic and conformational behavior may affect both interactions with drug delivery systems and with biological targets—thus leading to varied pharmacological activity.

## 2. Results

### 2.1. Structure and Stationary Spectroscopy

Figure 1A presents the structure of the *cis*-enol form of the selected compound from the 1,3,4-thiadiazole group, featuring a 2,4-dihydroxyphenyl substituent—a substituent commonly used in this series of compounds investigated by our group. Figure 1B, in turn, shows the structure of this compound in its excited-state keto tautomeric form*, which is favored in nonpolar environments. In Figure 1C,D, we present example aggregated forms of this molecule, several of which were considered in our earlier studies on the ESIPT phenomenon in solvent systems [22,23], Figure 1C shows *card-pack* type aggregates, while Figure 1D shows *head-to-tail* aggregates. The studied compound contains a characteristic 1,3,4-thiadiazole ring, which is connected via a –CH_2_- group to a naphthalene moiety. The molecular architecture—particularly the presence of naphthalene as a substituent—appears to influence the compound’s polarity, possible phase separations, and, most importantly, its propensity for π–π stacking-type aggregation [38].

Figure 2 presents example electronic absorption spectra (Figure 2A) and the corresponding fluorescence emission spectra for NTBD upon excitation at the main absorption maximum, recorded in several selected solvent systems. Referring to our earlier studies, Figure 2A shows three typical electronic absorption spectra for this compound, with the main absorption band maximum at ~325 nm, which—according to well-established literature standards—corresponds to a π → π* electronic transition within the molecular chromophore system. As can be seen, depending on the medium, the main band undergoes a slight solvatochromic shift.

For the aqueous system, however, the main band shifts slightly toward the bathochromic side, and in addition, a characteristic long-wavelength shoulder appears. According to M. Kasha’s exciton splitting theory, this indicates the formation of molecular aggregates, primarily head-to-tail dimers [39]. Far more intriguing effects are shown in Figure 2B. In a polar solvent, a single fluorescence emission band is observed, whereas in a 50:50 *v/v* water–methanol mixture, a distinct dual fluorescence effect appears—associated with excited-state intramolecular proton transfer (ESIPT), which we have discussed in detail in our earlier work for media such as various solvents and solutions with variable pH. The phenomenon lies in the fact that, following a single excitation, it is possible to observe two separate fluorescence emission bands [40,41]. The NTBD spectrum in water confirms both our results and those of other groups: by appropriately influencing the molecule’s tendency to aggregate, the ESIPT effect can be induced, which is often linked to aggregation-induced emission (AIE) and its associated fluorescence enhancement (AIEE) [37]. For the aqueous solution used here, we induced dual fluorescence despite the fact that the compound, when dissolved in MeOH, does not exhibit this effect. Therefore, in this study, we decided to investigate how the NTBD molecule behaves in liposomal systems formed from DPPC, which are increasingly used in biophysical studies, for example as effective drug carriers.

To better understand the precise molecular mechanisms occurring in liposomal systems containing NTBD, Figure 1 presents the electronic absorption spectra of the compound at various molar concentrations in DPPC: 1, 3, 5, 10, 15, and 20% mol NTBD in DPPC. As can be observed, for all concentrations prepared for measurement in the liposomal system, the main absorption maximum of the compound—known from solvent studies—appears at 325 nm [22] (Figure 2). Depending on the NTBD concentration, slight shifts in this band are observed, corresponding—according to well-established literature standards and our previous studies—to a π → π* electronic transition [22]. However, these solvatochromic shifts are small, amounting to only a few nanometers. In addition, on the long-wavelength side, at around 370 nm, a noticeable shoulder is observed on the main absorption band. This shoulder remains visible despite sample scattering, which is characteristic of this type of liposomal system.

It is therefore possible that the studied samples contain molecular forms other than monomers, such as dimers or larger N-aggregate systems. This is indicated both by the position of the band—on the long-wavelength side relative to the main absorption maximum—and by its noticeably lower intensity, a feature often observed in this type of sample.

Referring to the relationship described in M. Kasha’s exciton splitting theory:(1)$$R_{\beta }=\sqrt[{3}]{\frac{\mu^{2}\kappa }{\eta^{2}\beta }}$$
where *μ* is the transition dipole moment of the interacting molecules, *η* is the refractive index, and *β* is the point dipole–dipole interaction energy. According to M. Kasha’s exciton splitting theory, *κ* = 1 corresponds to a *card-pack* arrangement of molecules in an aggregate, while *κ* = −2 corresponds to a *head-to-tail* aggregate.

It is therefore possible to estimate the intermolecular distance between NTBD molecules in a dimer arranged in the *card-pack* configuration. In our case, this distance is approximately ~3.75 Å, which is generally consistent with our earlier calculations in solvent systems. The slight difference may result from the increased hydrophobicity of the lipid environment and the additional possible interactions between molecules forming the dimer. Particular attention should be given to the effect presented in Figure 2, which shows the fluorescence emission spectra corresponding to the electronic absorption spectra from Figure 1. The excitation wavelength for all samples was set to the main absorption maximum at 325 nm. The fluorescence emission spectra for 285 nm excitation are presented in the Appendix A. As can be seen, depending on the concentration of NTBD in the liposomal system, a phenomenon of dual fluorescence is observed. For 1% mol NTBD in DPPC, a single fluorescence band with a maximum at ~380 nm is observed, with a slight shoulder around 485 nm. As the NTBD content in the liposomal DPPC system increases, for example to 3% mol, in addition to the band at 380 nm, a long-wavelength emission appears with a maximum around 490 nm. The intensity of this long-wavelength band increases significantly with increasing NTBD concentration, while the intensity of the ~380 nm band simultaneously decreases. As shown in the inset of Figure 2, the emission ratio at 380 nm/490 nm changes from 4.98 to 0.24. Excitation at the wavelength corresponding to absorption by the strongly lipophilic naphthalene moiety, which significantly affects the molecule’s membrane penetration in the given biophysical system, produces a very similar effect, as shown in Appendix A.

In summary, these studies on NTBD in a model liposomal system confirm, based on our previous research, that the observed effects in the fluorescence emission spectra are related to the excited-state proton transfer (ESIPT) phenomenon [22,23]. What is particularly interesting, and will be discussed in detail in the following sections, is that in this case the ESIPT effect is strongly induced by molecular aggregation, which is in turn influenced by changes in the hydrophobicity of the lipid membrane. This aggregation significantly enhances the emission from the NTBD keto tautomer at 490 nm, as observed in this model biological system, which is important both from a biophysical standpoint and due to the increasing use of liposomes as drug delivery systems. This aspect is particularly relevant in light of recent literature, as depending on the molecular concentration, different molecular forms of NTBD are present, which in turn influence the energetics and therefore the pharmacological and biological potential of the molecule. It is also worth noting that during the preparation of these liposomal systems, NTBD was dissolved in an environment in which it typically exists in the enol tautomer form in various conformations, such as *cis*- or *trans*-. However, the relatively high hydrophobicity of the lipid environment promotes—or more accurately facilitates—aggregation interactions, which significantly lowers the potential barrier between the keto/enol transformation and thus enables the ESIPT process [36]. This was extensively studied in our previous work on solvent systems. Furthermore, many research groups have discussed the coexistence of ESIPT with aggregation-induced emission (AIE) in various molecules, not limited to thiadiazole-based systems [37].

Proceeding to more detailed considerations regarding the impact of aggregation effects on the observed ESIPT phenomenon in liposomal systems, enhanced by the NTBD molecular structure through AIE (Aggregation-Induced Emission) effects, Figure 3 presents the corresponding fluorescence excitation (Ex) spectra matching the results from Figure 1 and Figure 2. In Figure 3A, the fluorescence excitation spectra were obtained using an excitation wavelength corresponding to the first maximum in the fluorescence emission spectra, i.e., 380 nm. In Figure 3B, excitation spectra were recorded for the same samples, but emission was collected at the wavelength corresponding to the maximum of the keto* form of NTBD. Compared to absorption spectra, fluorescence excitation spectra generally exhibit much greater selectivity, especially in systems where more than one molecular form of the fluorophore is present. This means that, depending on the excitation wavelength, we can selectively excite either the monomeric form or various aggregated forms, such as dimers or N-aggregates. This is of critical importance when studying fluorophores exhibiting AIE-enhanced fluorescence [38]. In the first panel of Figure 3, we generally observe bands corresponding to the absorption spectra from Figure 1, but the main maximum here is significantly broadened compared to the respective absorption spectra, confirming the presence of aggregation-related processes and an increased likelihood of non-monomeric forms of the compound. However, the more interesting and compelling evidence for the influence of aggregation on the observed spectral effects related to ESIPT in the liposomal system comes from Figure 3B. Here, we see very strong aggregation even at the initial concentrations of 1, 3, and 5% mol NTBD in DPPC, with the main band still exhibiting a maximum around 325–330 nm. However, for NTBD concentrations of 10, 15, and 20% mol in DPPC, a strong aggregation band emerges, eventually shifting to ~340 nm. Additionally, from the long-wavelength side, we observe a broadening of the main band’s maximum in the ~370 nm region from the very beginning. These effects confirm a very strong influence of aggregation in the studied biophysical samples, the presence of which has a decisive impact on the induction of the ESIPT phenomenon in the considered 1,3,4-thiadiazole derivative. Subtly, though the image is somewhat blurred, we also see changes in the short-wavelength bands, further supporting the fact that from the very beginning, the system may contain both “card pack” and “head-to-tail” aggregated forms, known from exciton splitting theory, with the latter likely being more abundant [39]. Of course, at higher NTBD concentrations—shown here mainly to highlight the photophysical effect—other aggregated forms, such as X-aggregates, must also appear, eventually leading to more disordered structures with increasing chromophore concentration [8]. This rapidly leads to an increase in long-wavelength fluorescence emission associated with the keto* tautomeric form of NTBD, which is precisely what we observe in Figure 2 with increasing NTBD concentration in the liposomal system.

In summary, this part of the study demonstrates a significant influence of aggregation on the excitation spectra, which directly translates into the observed spectral effects in the fluorescence emission spectra. The ESIPT process is strongly enhanced here by aggregation-induced effects (AIE), further facilitated by the amphipathic nature of the chosen liposomal environment.

As confirmed by our earlier research, aggregation in this system must drastically lower the energy barrier for the enol ↔ keto tautomeric transition, thereby promoting the ESIPT mechanism.

A strong confirmation of the aggregation effects associated with AIE in NTBD incorporated into DPPC liposomal systems was further provided by measurements of resonant light scattering (RLS) spectra, presented in Figure 4. The presence of RLS bands, in line with the findings of Pasternack et al., can be attributed to chromophore aggregation of the studied molecules [39]. In this work, as well as in numerous other literature reports, it has been shown that the RLS signal intensity increases exponentially with the number of chromophores in the system. As can be seen in Figure 4, the corresponding synchronous spectra related to the results from Figure 1, Figure 2 and Figure 3 are presented.

A clear dependence of the RLS signal on the NTBD concentration in the liposomal system was observed. Comparing the RLS intensity for 1 mol% NTBD in DPPC with that for 20 mol% NTBD in DPPC, we see that the signal intensity around 350 nm nearly doubled. Interestingly, at lower concentrations, the intensity initially decreased, suggesting that larger aggregated structures may have begun forming at the expense of smaller assemblies. Moreover, the oscillatory pattern of the RLS bands may indicate the coexistence of multiple aggregated forms of the molecule within the liposomal system. Overall, the obtained RLS spectra clearly corroborate the fluorescence excitation results discussed above.

The next part of the discussion pertains to calculations and analyses of the quantum yield of fluorescence for NTBD incorporated into DPPC liposomes at different molar fractions (5–20%) to quantify the enhancement of ESIPT in aggregated states. The results are presented in Table 1. Solutions of the reference fluorophore Coumarin1 were prepared at concentrations matching those of NTBD used in the liposomal samples shown in Figure 1, and their fluorescence spectra were subsequently recorded (Appendix A).

With increasing NTBD content in the lipid bilayer, a notable increase in emission intensity is observed, reflecting stronger aggregation inside the liposomal matrix. Simultaneously, the fluorescence quantum yield gradually increases from 0.19 at 5 mol% NTBD to 0.31 at 20 mol%. This is fully consistent with the AIE (Aggregation-Induced Emission) mechanism, where restricted intramolecular motions in aggregated environments suppress non-radiative deactivation pathways. Therefore, the progressive rise in fluorescence quantum yield directly demonstrates that aggregation promotes the ESIPT process for NTBD in liposomes.

Summarizing the spectroscopic data obtained in the liposomal environment, it is evident that in the aggregated state, NTBD molecules become significantly more rigid due to the restriction of their internal motions (rotational and vibrational). This intramolecular restriction (RIR/RIV) effectively suppresses non-radiative relaxation pathways (e.g., the TICT state), thereby maintaining the molecule in a configuration favorable for intramolecular proton transfer [42]. As a result, the energy barrier for the enol–keto tautomeric transition is lowered, which enhances the emission of the keto form (increased ESIPT contribution). This observation is consistent with the RIM theory and the experimentally observed increase in fluorescence quantum yield at higher NTBD concentrations within the studied system. It should therefore be emphasized that molecular aggregation reduces the potential energy barrier associated with the tautomeric transition, facilitating the occurrence and detection of the ESIPT process even in environments markedly different from nonpolar solvents, where this phenomenon is more commonly observed.

As a complement to the synchronous spectra measurements, we decided to include the results of steady-state fluorescence anisotropy measurements, which are directly related to the spectral measurements in polarized light. In this study, fluorescence anisotropy was used to assess the influence of the liposomal system on the aggregation of NTBD molecules and to interpret the stabilization or destabilization of the gel or fluid phase of DPPC lipids. The results are presented in Figure 5 for the liposomal system at two selected temperatures corresponding to different lipid phases: 22 °C—gel phase (L_β′_) below the phase transition, and 42 °C—liquid-crystalline, fluid phase (L_α_) above the main phase transition of the lipid [43]. When DPPC liposomes are in the gel phase (L_β′_), they exhibit high anisotropy due to the rigid, ordered arrangement of the lipids. As the temperature increases, DPPC undergoes a main phase transition from the gel to a disordered (liquid-crystalline) phase, and the system’s anisotropy usually decreases [44]. An interesting effect is shown in Figure 5A for 5 mol% NTBD in the 440–480 nm range, where a sharp increase in steady-state anisotropy is observed at 42 °C (r = 0.35 at 450 nm) compared to 22 °C (r = 0.21 at 450 nm). Above the main phase transition, the system exhibits a low degree of acyl chain packing and low viscosity, which allows the formation of NTBD aggregates and stabilizes and rigidifies the lipid membrane. Above 480 nm, the anisotropy values are similar for both temperatures. Figure 5B presents the results for a higher NTBD concentration (10% mol) in the same system. A decrease in r compared to the lower NTBD concentration is observed, which may indicate higher mobility of NTBD molecules due to the presence of both aggregates and monomers in the lipid bilayer. At the higher thiazole concentration, higher anisotropy is observed in the liquid phase (L_α_) than in the gel phase (L_β′_), confirming the ability of NTBD to aggregate in the lipid environment. The temperature increase that relaxes the DPPC acyl chains thus facilitates NTBD aggregate formation and stabilizes the lipid membrane. It is also worth noting the pronounced change in anisotropy at shorter wavelengths. Assuming that the blue and red ends of the spectrum correspond to different subchromophores, the observed results may indicate changes in the molecular dynamics of NTBD [45]. It seems more likely, however, that the lifetime of the short-wavelength fluorescence component is changing. If it has shortened—which is clearly indicated by the TCSPC data (as described later in the work) showing a shorter lifetime—then the observed increase in anisotropy is a direct consequence of this. On the other hand, if the long-wavelength component is responsible for the longer lifetime and has only slightly shortened, then the lack of changes in anisotropy in this range is understandable. The interpretation of fluorescence anisotropy provides complex information concerning both the orientation and the mobility of the fluorescent probe within the lipid bilayer.

The next stage of the study involved measuring fluorescence lifetimes using the TCSPC technique in a DPPC liposomal system with the addition of NTBD at various molar concentrations (1, 3, 5, 10, 15, 20% mol) at room temperature (T = 22 °C). Excitation was induced with light of 372 nm wavelength. The decays could be well fitted with biexponential models; simpler models were insufficient, while more complex models did not significantly improve the fitting quality. Typical fluorescence decay profiles for selected NTBD concentrations (1, 5, 20% mol) are illustrated in Figure 6. The results of the fluorescence lifetime analysis are presented in Table 2. It was observed that, with increasing NTBD concentration in the DPPC lipid system, the average fluorescence lifetimes slightly increased (from 1.38 ns to 1.61 ns). In this case, referring to the results presented earlier in this article, it can be inferred that in the forming liposomal system, which is in the gel phase (L_β′_), continuous aggregation of NTBD molecules may occur.

Additionally, referring to our previous studies of NTBD molecules in various polar/nonpolar environments [22], time-domain measurements showed that two significantly different fluorescence lifetime components could be clearly identified. The short-lived component was characterized by a lifetime of approximately 0.6 ns (0.55–0.76 ns), while the long-lived component was around 2 ns (1.98–2.19 ns). Time-resolved fluorescence spectroscopy is a highly sensitive and specific experimental method that facilitates the detection of different tautomeric forms of the fluorophore, for example, in a lipid environment. The observation of two fluorescence lifetime components thus confirms the coexistence of two distinct excited states of the NTBD molecule. As shown in Figure 2, depending on the concentration of NTBD in the liposomal system, the phenomenon of dual fluorescence is observed, which is reflected in the recording of two lifetime components. Even at a concentration of 1% mol NTBD, where single emission occurs, the observed decay is clearly biexponential. According to the literature [46], the excited *keto** form is associated with the shorter lifetime, whereas the longer lifetime corresponds to the excited *enol** tautomer. Additionally, it can be observed (subtly but noticeably) that the contribution of the first component decreases with increasing NTBD concentration, while that of the second component increases significantly. This may be related to the fact that, as the number of thiazole molecules increases, the likelihood of aggregation interactions between them also rises, causing the contribution of the *keto* tautomer to slightly diminish. Fluorescence lifetime measurements of NTBD in the liposomal system confirm the existence of the ESIPT effect as well as the interplay between ESIPT and aggregation effects.

All of the above studies from this series of measurements related to changes in NTBD concentration in the lipid system were complemented by dynamic light scattering (DLS) measurements, presented in Figure 7. These measurements were performed to compare the hydrodynamic size (Figure 7A) and polydispersity index PDI (Figure 7B) of DPPC liposome particles with the addition of NTBD at various molar concentrations (1, 3, 5, 10, 15, 20% mol) at room temperature. For DPPC alone, the average particle diameter exceeded 3000 nm, suggesting the presence of large multilamellar vesicles (MLVs) [47]. However, after adding NTBD, the average diameters were reduced to sizes of around ~1500 nm for a concentration of 15% mol. The presence of NTBD can disrupt the ordered structure of the bilayer through interactions with the lipid membrane, altering its flexibility and packing density. DPPC liposomes may also facilitate the solubilization of forming NTBD aggregates, which can lead to a reduction in the vesicle size. Moreover, the PDI ranged from 0.2 for 5 mol% NTBD up to 0.68 for 15 mol% NTBD (Figure 7B and Appendix A), indicating moderate uniformity of the liposome dispersions and confirming that NTBD incorporation did not significantly affect the overall stability of the liposomal vesicles.

### 2.2. UV-Vis, Temperature-Dependent Fluorescence—Spectroscopic Measurements as a Function of Environmental Temperature Changes

In the next stage of the study, similar measurements were conducted in the liposomal system as a function of temperature changes, due to the fact that these changes reflect significant transformations within the liposomal system. The chosen lipid system was aimed at enhancing the ESIPT effect and the associated dual fluorescence phenomenon, so that aggregation processes (observable as AIE) could be facilitated. Figure 8A,B show the electronic absorption spectra of NTBD at two molar concentrations in DPPC as a function of the temperature of the analyzed medium (5% mol NTBD in Figure 8A and 10% mol in Figure 8B). Results for a higher concentration of 20% mol NTBD in DPPC are presented in Appendix A. As can be observed, similar to Figure 1, all absorption spectra exhibit a broad band extending from approximately 300 to 350 nm with a maximum at 325 nm, characteristic of the π → π* electronic transition of the analyzed molecule [22]. Depending on the temperature, a noticeable decrease in the intensity of the main absorption band occurs. At the same time, NTBD shows a very pronounced broadening of bands on the long-wavelength side, with a wide maximum around 370 nm, indicating the presence of strong aggregation effects, as previously described in the main text. The position of this long-wavelength band and its significantly lower intensity clearly suggest the likely presence of forms other than monomeric species in the analyzed samples (dimers or N-aggregates). Additionally, it can be noted that on the long-wavelength side, as the temperature increases and approaches the main phase transition, the band intensity begins to rise slightly. As the lipid undergoes the main phase transition (from the L_β′_ to L_α_ phase), the membrane fluidity changes, facilitating mutual interactions between the NTBD molecules.

Particularly interesting effects are observed in the fluorescence emission spectra of NTBD in Figure 9 and Appendix A, corresponding to the respective electronic absorption spectra from Figure 8 and Appendix A. The excitation wavelength for all samples corresponds to the absorption maximum at 325 nm. As shown in Figure 9A,B and Appendix A, the phenomenon of dual fluorescence is observed, with maxima at ~380 nm and 480/500 nm. Depending on the NTBD concentration in the liposomal system and the system temperature, different spectral bands appear. The intensity of the long-wavelength band decreases with increasing temperature, while the intensity of the short-wavelength band with a maximum at ~380 nm simultaneously increases significantly.

As presented in the inset of Figure 9A, the emission ratio at 380 nm/490 nm changes from 0.5 to 7 for the lowest concentration of 5% mol, whereas for higher concentrations the ratio changes less significantly, ranging from 0.1 to 0.7. Notably, Figure 9A, showing the fluorescence emission spectra for 5% mol NTBD in DPPC, is particularly interesting. In the initial gel phase (L_β′_), the band at ~500 nm—associated, according to our previous studies on NTBD, with the emission of the *keto** tautomer—is the dominant form.

However, with increasing temperature and approaching the pre-transition temperature of ~35 °C for DPPC, there is a significant increase in the intensity of the band with a maximum at 380 nm (characteristic of monomeric enol forms, both *cis*- and *trans*- conformations) and a disappearance of the long-wavelength emission.

For higher NTBD concentrations (10% mol and 20% mol), as the system transitions from the L_β′_ → L_α_ phase, the band at ~490 nm also decreases, accompanied by an increase in the intensity of the band at 380 nm. However, in these systems, the dominant form remains the 490 nm band characteristic of the ESIPT effect in NTBD, corresponding to the *keto** tautomer. This effect suggests the presence of different forms of molecular organization in the membrane, depending on concentration and temperature.

As indicated in previous studies, these effects in the examined molecule are likely related to molecular aggregation occurring in model biological liposomal systems. That is, there is a continuous shifting of the tautomeric equilibrium between the *keto** and *enol** forms; however, with increasing NTBD concentration, it is evident that aggregation interactions support the existence of emission from the *keto** tautomer (Figure 9 and Appendix A). Further evidence of the significant influence of aggregation on the excited-state intramolecular proton transfer (ESIPT) phenomenon will be presented in the following part of the study.

Appendix A shows the fluorescence excitation (Ex) spectra for NTBD, corresponding to the fluorescence emission spectra in Figure 9B and Appendix A. The excitation emission was recorded at wavelengths corresponding to the maxima of the respective fluorescence emission spectra (380 and 480 nm). In Appendix A, the fluorescence excitation spectra were obtained using excitation at the wavelength corresponding to the first maximum of the emission spectra, i.e., at 380 nm. In Appendix A, for the same samples, spectra were recorded where the emission upon excitation was collected at the wavelength corresponding to the maximum of the *keto** form of NTBD, i.e., the long-wavelength emission in the fluorescence spectrum.

The much higher selectivity of the fluorescence excitation spectra compared to electronic absorption spectra facilitates excitation of a specific molecular form of the compound in the given lipid system, such as monomer, dimer, or a larger aggregated assembly. When excited at the wavelength corresponding to the first maximum of the emission spectra (380 nm), Appendix A show that the main maximum is significantly broader compared to the corresponding absorption spectra, and with increasing temperature, it rises considerably and shifts from 325 nm to 320 nm. In contrast, upon excitation at Ex 480 nm, the main maximum decreases with increasing temperature and slightly shifts toward longer wavelengths, from 330 nm to 335 nm, confirming the presence of molecular aggregation processes. It is also worth noting that all measured excitation spectra in Appendix A are very broad, further confirming the growth of aggregated NTBD molecular assemblies.

In Figure 10, fluorescence excitation (Ex) spectra for NTBD at a concentration of 10% mol are shown at 22 °C (Figure 10A) and 42 °C (Figure 10B). Evidence for the influence of specific types of aggregation (capable of inducing the predominance of a given type of fluorescence effect) is presented in Figure 10, where the normalized fluorescence excitation spectra are overlaid on the 1-T (T-transmittance) spectra, with additional differential spectra for each case shown in the lower schematics under Figure 10A,B. The NTBD spectra shown in Figure 10A,B, under short-wavelength excitation, exhibit a significant predominance of “card pack” type aggregation, as described in exciton splitting theory [39]. Of course, it should be noted that this type of aggregation may dominate only in the initial stage of the process; at significantly higher concentrations, other types of aggregated forms, as reported in the aforementioned work by M. Kashy, naturally appear.

In the next stage of our study, resonance light scattering (RLS) spectra of NTBD in the DPPC lipid system were recorded, which—according to Pasternack—are associated with aggregation of the fluorophore chromophores, as mentioned above. Figure 11 presents the RLS spectra for 5% mol (Figure 11A) and 10% mol (Figure 11B) NTBD in the liposomal system as a function of temperature. In the Appendix A, results for the highest NTBD concentration, 20% mol, are shown.

The intensity of the RLS spectra decreases with increasing temperature of the system, confirming that the number of aggregated forms decreases while the number of monomeric forms increases with temperature. Nevertheless, the RLS spectra in the studied lipid system remain fairly intense, even at the lowest NTBD concentration (5% mol), indicating the presence of aggregated forms and their influence on the observed effects.

Additionally, Figure 11C shows the dependence of absorbance at 325 nm on temperature for 5% and 10% mol NTBD in DPPC. In this case, a relatively significant decrease in spectral intensity can be observed, particularly as the system approaches the main phase transition of DPPC (~41 °C).

### 2.3. Temperature Dependence of Fluorescence Lifetimes

The next stage of the study involved measuring the fluorescence lifetimes of NTBD in the DPPC liposomal system as a function of temperature. As before, the technique of time-correlated single-photon counting (TCSPC) was employed. Measurements were performed over a temperature range from 20 °C to 55 °C for systems with different molar concentrations of NTBD (5, 10, 20% mol). Excitation was induced using light at a wavelength of 372 nm. The fluorescence lifetime results are presented in Table 3.

Typical fluorescence decay profiles for selected NTBD concentrations (5, 10, 20% mol) are illustrated in Figure 12 and Appendix A. As previously, the obtained data could be satisfactorily fitted using a biexponential model. In all cases, two fluorescence lifetime components could be identified, confirming the coexistence of two different emitting states of NTBD, as described above. According to the literature [48], the excited *keto** form is characterized by a shorter lifetime (τ_1_), whereas a longer fluorescence lifetime (τ_2_) is typical for the excited *enol** tautomer.

For all studied concentrations, a decrease in the average fluorescence lifetime of NTBD was observed with increasing temperature, as shown in Appendix A. In all cases, the changes in average lifetimes exhibited a similar temperature dependence. Increasing the temperature for all tested concentrations caused a significant reduction in the shorter lifetime component (by 70–88%) and a smaller reduction in the longer lifetime component (by 3.5–15%). However, the key contribution to the shortening of the average fluorescence lifetime is the change in the relative contributions of the individual components. An increase in temperature is accompanied by an increase in the contribution of the short-lived component and a decrease in the contribution of the long-lived component. It is worth noting that although these changes are monotonic, the greatest dynamics were observed over a narrow temperature range characteristic of the gel-to-liquid crystalline phase transition (L_β′_ → L_α_).

Appendix A presents the change in the contributions of the first and second components for 10% mol NTBD in the DPPC lipid system. The presented results clearly indicate the presence of different oligomeric forms of the fluorophores, whose proportions change depending on the temperature and the phase of the liposome. It should be emphasized that the biexponential fluorescence lifetime clearly suggests a relationship between the dual fluorescence effect of the NTBD molecule in the studied DPPC liposomal system and molecular aggregation. More precisely, it points to the influence of aggregation on the emission from the *keto** form, specifically its role in lowering the energy barrier for the *keto** ↔ *enol** tautomeric transition [10].

### 2.4. Infrared Spectroscopy—FTIR in the Analysis of Conformational Changes in the Lipid Membrane Induced by NTBD

To investigate the molecular interactions occurring between the 1,3,4-thiadiazole compound and the DPPC membrane, FTIR spectroscopy was employed. To facilitate the interpretation of the results, Figure 13 shows correlations between the positions of absorption maxima of selected bands and temperature in the range of 22–46 °C for pure DPPC and for NTBD at concentrations of 3% mol and 5% mol in the lipid system.

#### 2.4.1. Hydrophilic Region of the Membrane


**ν(C=O)**


The effect of NTBD addition on changes associated with the stretching vibrations of the carbonyl (C=O) group present in the lipid structure is shown in Figure 13A. As the temperature increases up to the pre-transition phase of the lipid (~35 °C), the carbonyl stretching region, centered at ~1733 cm^−1^, practically overlaps for both NTBD and DPPC. As the system approaches the main phase transition, a sudden shift in the band maximum toward higher frequencies is observed, especially at higher NTBD concentrations. For 5% mol NTBD, the band shifts to ~1737 cm^−1^ at temperatures above the main phase transition. The shift in the carbonyl stretching region to higher frequencies with increasing NTBD concentration confirms the interaction between NTBD molecules and the C=O group of the lipid through hydrogen bonding.


**ν_as_(PO_2_)**


Simultaneously, changes can be observed in the region centered at ~1240 cm^−1^, characteristic of the asymmetric phosphate (PO_2_^−^) skeletal vibrations in DPPC (Figure 13F). NTBD molecules at 3% and 5% mol interact with the phosphate group of the lipid, which is manifested as a slight shift toward higher frequencies (1243 cm^−1^) already at room temperature. Moreover, comparing the results for pure lipid and lipid with NTBD, it can be seen that NTBD stabilizes the membrane, as indicated by the absence of fluctuations in vibrations below the lipid phase transition temperature. At the main phase transition of DPPC, a small shift in the vibrations toward higher values (~1244 cm^−1^) is also observed for the studied system. These results suggest that NTBD molecules at 3% and 5% mol are only partially accumulated in the hydrophilic region of the lipid headgroups and exhibit weak interactions with the −PO_2_^−^ group.


**ν(C–O–P–O–C)**


Another characteristic region in DPPC where changes can be observed is the skeletal vibrations of C–O–P–O–C at ~1087 cm^−1^ (Figure 13C). At 3% mol NTBD in DPPC, changes are relatively minor both before and after the main phase transition L_β′_ → L_α_, with a slight shift in the vibrations toward higher frequencies (~1088 cm^−1^). Changes are more pronounced at higher NTBD concentrations (5% mol) in DPPC. The characteristic band for C–O–P–O–C vibrations, with a maximum around 1088 cm^−1^, is clearly shifted to higher frequencies at the main phase transition temperature of DPPC. These results indicate interactions between NTBD molecules and the hydrophilic regions of the lipids.


**(–N^+^(CH_3_)_3_)**


Changes in the region corresponding to the choline group (–N^+^(CH_3_)_3_) are shown in Figure 13G. The band shifts toward higher wavenumbers from 965 cm^−1^ in DPPC to 969 cm^−1^ in NTBD/DPPC, while no temperature-dependent shift is observed. This indicates that NTBD at these concentrations may stabilize the lipid membrane, as the DPPC phase transition is diminished. Around 43 °C, for 5% mol NTBD, a small shift in vibrations toward lower wavenumbers occurs, which may suggest the presence of some free NTBD molecules resulting from the disaggregation of the system at higher temperatures, which may interact differently with the lipid membrane.

#### 2.4.2. Hydrophobic Region of the Membrane


**δ(CH_2_) + δ(CH_3_)**


For the studied lipid DPPC, the bending vibrations of the –CH_2_ groups are located around ~1340 cm^−1^, while the bending vibrations of the –CH_3_ groups are observed around ~1374 cm^−1^ (Figure 13B,E). Incorporation of NTBD molecules into DPPC at concentrations of 3% and 5% mol slightly stabilizes the membrane, causing a small shift in the –CH_3_ bending vibrations toward higher wavenumbers, and no changes in band positions with increasing temperature.

For the –CH_2_ bending vibrations centered at ~1343 cm^−1^, minor shifts toward lower wavenumbers are observed at 3% mol NTBD in DPPC, whereas more significant changes in band position occur at 5% mol NTBD. These shifts display opposite trends both before and after the main phase transition of the lipid.


**ν_s_(CH_2_) + ν_s_(CH_3_)**


Figure 13D,H show the effect of NTBD on the stretching vibrations of the –CH_2_ and –CH_3_ groups, observed between 2800 and 3000 cm^−1^. As illustrated, NTBD at 3% and 5% mol shifts the –CH_2_ and –CH_3_ stretching vibrations to higher wavenumbers, while no temperature-dependent shift is observed. This indicates that NTBD may stabilize the lipid bilayer, as the phase transition in DPPC is suppressed.

### 2.5. XRD of Multilayer

Appendix A presents X-ray diffraction data for DPPC multilayers containing NTBD at concentrations of 5%, 10%, and 20% mol. Analysis of the obtained diffractograms indicates clear changes in both the positions and intensities of the main Bragg reflections, reflecting the impact of NTBD on the lamellar ordering and molecular organization of the lipid multilayers.

For lower NTBD concentrations (5–10% mol), a slight shift in the main reflection toward higher 2θ angles was observed, which may indicate an increase in the interlayer spacing (d-spacing), likely resulting from changes in the hydration of the structures or partial intercalation of NTBD molecules between the lipid bilayers. At the same time, a moderate increase in the reflection intensity is noticeable, which can be interpreted as an enhancement of structural order and coherence in the sample. This effect may arise from stabilizing interactions between NTBD and the polar groups of the lipids (e.g., hydrogen bonding) or π–π interactions with aromatic moieties.

For the highest concentration (20% mol), a significant decrease in the intensity of the main reflection is observed, returning to the baseline level, indicating a disruption of the multilayer ordering. This is likely due to a saturation effect, where an excess of NTBD leads to its partial aggregation or phase segregation, limiting its effective incorporation into the bilayer structure. Such an effect has been previously reported in the literature when studying the influence of various amphiphilic or hydrophobic compounds on phospholipid structures, where exceeding a certain concentration threshold results in destabilization of lamellar systems and even transitions to non-lamellar structures [49,50]. Analyzing the reflections at higher diffraction angles, a characteristic intensity profile is clearly visible—an initial increase followed by a decrease. This trend confirms the overall effect of NTBD on the structural ordering across the entire range of examined reflections. The nonlinear dependence of 2θ values on NTBD concentration (Panel B) suggests a complex response of the system to the introduced compound—at lower concentrations, stabilizing interactions that reinforce the structure dominate, whereas at higher concentrations, disruptive effects prevail, leading to reduced ordering.

In summary, the X-ray diffraction results clearly indicate that NTBD interacts with the structure of lipid multilayers in a concentration-dependent manner. At moderate addition levels, an increase in ordering and layer expansion is observed, whereas at higher levels, saturation effects and partial structural destabilization occur. The obtained results are consistent with existing knowledge on the influence of small-molecule compounds on the organization of phospholipid membranes and may also be relevant for understanding the mechanisms of NTBD’s action as a potentially bioactive additive or carrier in model membrane systems.

### 2.6. Single Crystal X-Ray Diffraction

Ultimately, we decided to present a crystallographic analysis of a selected molecule, which clearly illustrates the formation of dimeric arrangements and larger aggregated structures. Crystallographic studies make it possible to obtain detailed information about the molecular structure and potential molecular interactions, providing a basis for understanding their behavior in a lipid environment. Combining crystallographic and spectroscopic data allows for the assessment of changes in the dynamics and ordering of lipid membranes in the presence of the studied thiazole-based molecules.

Structural studies confirm the molecular structure of NTBD (Appendix A). The resorcinol and thiadiazole ring fragments are similar to those of analogous compounds [51,52]. The hydroxyl groups O1-H2 and O2-H2 in the crystal form a network of hydrogen bonds, acting both as donors and acceptors. The nitrogen atoms in the thiadiazole ring serve only as hydrogen bond acceptors. Specifically, the N1 atom forms an intermolecular N1^…^H1–O1 hydrogen bond (where H1 and O1 come from a neighboring molecule), while another nitrogen atom forms an intramolecular bond with the hydrogen atom H2 (Appendix A). The presence of the naphthalene group gives the NTBD molecule an amphiphilic character. In the crystal, the resorcinol and thiadiazole rings form the hydrophilic layer, whereas the naphthalene and methylene groups form the hydrophobic layer (Appendix A). This property is particularly significant for the incorporation of NTBD into lipid membranes. In this structure, π-stacking is not observed, which is related to the specific arrangement of NTBD molecules and the nonlinear architecture of the compound.

## 3. Materials and Methods

The synthesis of NTBD was carried out at the Department of Chemistry, University of Life Sciences in Lublin. The detailed process was described in a previous publication [53].

### 3.1. Methods

The phospholipid dipalmitoylphosphatidylcholine (DPPC) with >99% purity was purchased from Avanti Polar Lipids (Alabaster, AL, USA). Liposomes were prepared by thin film hydration at concentrations appropriate for the specific test method. NTBD was dissolved in methanol, and DPPC lipid in chloroform. The mixtures were prepared in glass tubes and evaporated under constant stirring under a nitrogen stream until a dry film formed. Additionally, the samples were evaporated under a vacuum of 10^-5^ bar for 1 h (using an AGA LABOR vacuum pump, AGA Systems, Centerville, UT, USA). The samples were then flooded with PBS buffer at 45 °C and left in a bath for 10 min (Polsonic, Sonic-6, Poznań, Poland). The samples were then disintegrated in an ultrasonic bath for 1 min and vortexed for 1 min (VELP Scientifica, Deer Park, NY, USA), the cycle repeated five times. Sonication was then performed using a Techpan UD-20 ultrasonic disintegrator for 3 × 3 s at 45 °C and 22 kHz. In this way, liposomes with a diameter of 90 to 110 nm were obtained.

### 3.2. Measurements of Electronic Absorption and Fluorescence Spectra

A Cary 300 Bio dual-beam UV-Vis spectrophotometer (Varian, Palo Alto, CA, USA) was used to measure the electronic absorption spectra of NTBD in liposomal systems. Temperature measurements on the loaded sample were made possible by a thermocouple probe with a thermostated cuvette holder and a 6 × 6 multi-cell Peltier block (Varian Cary Series II).

A Cary Eclipse spectrofluorometer (Varian) was used to record fluorescence excitation, emission, and synchronous spectra at room temperature. Fluorescence spectra were recorded with a resolution of 0.5 nm and corrected for the spectral characteristics of the lamp and photomultiplier. Resonance light scattering spectra were measured as suggested by Pasternack and Collings. The excitation and emission monochromators of the spectrofluorometer were scanned synchronously (i.e., without a gap between the excitation and emission wavelengths). The slits were then adjusted to achieve a spectral resolution of 1.5 nm. Analysis of the obtained spectra was performed using Grams/AI 8.0 software (Thermo Electron Corporation, Waltham, MA, USA) and Origin 2023b software (USA).

### 3.3. Anisotropy Measurements

The polarized components of fluorescence emission were used to calculate the ground-state fluorescence anisotropy (*r*) using the following formula:(2)$$r=\frac{I_{VV}-GI_{VH}}{I_{VV}+2GI_{VH}}$$
where *I_VH_* and *I_VV_* are the measured light emission intensities with horizontal and vertical polarization, respectively, recorded after excitation of the sample with vertically polarized light. *G* is a factor correcting for differences in the sensitivity of the detection system to light with vertical and horizontal polarizations. To enable UV light transmission, a grating polarizer was used for excitation.

### 3.4. Time-Correlated Single Photon Counting (TCSPC)

Time-correlated single photon counting (TCSPC) measurements were carried out using a FluoroCube fluorimeter (Horiba, Palaiseau, France). Samples were excited with a pulsed NanoLED diode operating at 372 nm (pulse duration: 150 ps) and a repetition rate of 1 MHz. To prevent pulse pile-up, the excitation intensity was adjusted using a neutral density gradient filter. Fluorescence emission was detected with a TBX-04 picosecond detector (IBH, Jobin Yvon, Middlesex, UK). Data acquisition and analysis were performed using DataStation and DAS6 software (version 6.4) (IBH, Jobin Yvon, UK).

All fluorescence decays were recorded in 10 × 10 mm quartz cuvettes, using a long-pass emission filter transmitting wavelengths above 408 nm. Excitation profiles required for deconvolution analysis were obtained in the absence of emission filters, using a light-scattering cuvette.

Measurements were performed for liposomal systems containing NTBD at 1, 3, 5, 10, 15, 20% mol in aqueous solution, at the various temperatures: room temperature (22 °C) and within the range of 20 °C to 55 °C. Each fluorescence decay was analyzed using a multiexponential model described by the following equation:(3)It=∑iαiexp(−tτi)
where *α_i_* and *τ_i_* are the pre-exponential factor and the decay time of *i*-th component, respectively. Best-fit parameters were obtained by minimizing the reduced *χ*^2^ value and analyzing the residuals of the experimental data. The fractional contribution (*f_i_*) of each decay time and the average lifetime of fluorescence lifetime (<*τ*>) were calculated according to the following equations:(4)$$f_{i}=\frac{\alpha_{i}\tau_{i}}{\sum_{j}\alpha_{j}\tau_{j}}$$(5)$$\langle \tau \rangle =\sum_{i}f_{i}\tau_{i}$$

### 3.5. DLS Methods

Dynamic light scattering (DLS) measurements were carried out at 25 °C using a Zetasizer Nano ZEN1600 (Malvern Instruments Ltd., Worcestershire, UK) equipped with a red (633 nm) laser. Data analysis was conducted using Zetasizer software version 7.13 (Malvern Instruments Ltd., Worcestershire, UK). Reported particle sizes are based on the number-weighted particle size distribution (PSD). For each sample, four measurements were performed, each consisting of six runs of 10 s duration. A detection angle of 173° was used for size determination.

### 3.6. FTIR Spectroscopy

Infrared FTIR spectra were measured using a Vertex70FTIR spectrometer (Bruker, Billerica, MA, USA). An attenuated total reflection (ATR) configuration was used with 10 internal reflections in a ZnS crystal slab (45° cut). For each spectral measurement, 24 scans were performed with a resolution of 2 cm^−1^. Spectra were collected while heating the samples from 23 °C to 38 °C in 3 °C increments and from 38 °C to 45 °C in 1 °C increments. All sample measurements were performed in the spectral range from 4000 to 400 cm^−1^. The instrument was continuously purged with N2 for 40 min both before and during the measurement. The ZnS crystal slab was cleaned using ultrapure organic solvents. Samples were prepared as described in the section above. Lipid samples were spread on a ZnS crystal surface and measured in liquid form. Analysis of the results was performed using Origin 2023b software (USA). All recorded spectral bands were shifted to the same background level before analysis. The concentrations used in the above measurements were 10 mg DPPC/1 mL PBS buffer and 3% and 5% mol NTBD in DPPC.

### 3.7. XRD

Diffraction data for NTBD were collected at 296 K using the CuKα radiation on a SuperNova (Oxford Diffraction, Oxfordshire, UK) diffractometer. Cell refinement, data collection as well as data reduction and analysis were performed with the CRYSALIS ver. 39.29d [22].

All structures were solved by direct methods [23] and refined using SHELXL [24]. The refinement was based on squared structure factors (F^2^). Non-hydrogenatoms were refined with anisotropic displacement parameters. Hydrogen atoms were located in the idealized geometric positions. The final crystallographic data for structural analysis are available from the Cambridge Crystallographic Data Centre (CCDC) under deposition number 2453047. These data can be obtained free of charge via www.ccdc.cam.ac.uk/structures, accessed on 28 October 2025 [54,55,56].

### 3.8. Fluorescence Quantum Yields

The quantum yields of fluorescence of NTBD liposomes with DPPC lipid were calculated and determined using 7-diethylamino-4-methylcoumarin (coumarin1) in methanol $\phi_{F}=0.59$ [57]. The final fluorescence quantum yield were calculated based on Equation (6):(6)$$\phi_{F(X)}=\phi_{R\left(MeOH\right)}(\frac{\lambda_{exR\left(MeOH\right)}}{\lambda_{exX}})(\frac{I_{X}}{I_{R\left(MeOH\right)}})(\frac{\eta_{X}^{2}}{\eta_{R(MeOH)}^{2}})$$
where the subscript *X* denotes corresponding NTBD liposomes at different concentrations, $\lambda_{ex}$ is the value of absorbance at the excitation wavelength, *I*—the plane under the emission curve, and *η* the refractive index of the solvent.

## 4. Conclusions

The spectroscopic studies presented in this work on the new 1,3,4-thiadiazole derivative, NTBD, provided comprehensive evidence linking the observed dual fluorescence in NTBD spectra primarily to the phenomenon of excited-state intramolecular proton transfer (ESIPT). Complementing our previous considerations in solvent systems, it is known that the higher-energy band originates from the *enol** tautomer, while the long-wavelength, lower-energy band corresponds to the *keto** tautomer.

Most importantly, from the perspective of this study, is the strong influence of aggregation-induced emission (AIE) effects on the induction of long-wavelength emission. The aggregation effects were enforced by the amphiphilic properties of the liposomal environment formed in this case from the phospholipid DPPC. Notably, the selected 1,3,4-thiadiazole was incorporated into the lipid environment after being dissolved in a polar medium, in which only fluorescence from the *enol** form is observed. Furthermore, the aggregation process occurring within the lipid bilayers leads to significant rigidification of the NTBD molecules, which, combined with their intrinsic molecular structure, enhances fluorescence intensity while effectively blocking non-radiative pathways of excited-state deactivation.

Infrared FTIR spectroscopy studies reveal the main sites of organization of the selected 1,3,4-thiadiazole within DPPC lipid membranes. Across the full temperature range affecting lipid membrane fluidity, analysis of absorption, fluorescence, RLS, and fluorescence anisotropy spectra shows that NTBD can exist in both monomeric and aggregated forms within the liposomal system. Upon exceeding the main phase transition temperature of the lipid, the aggregated forms can disaggregate, and monomeric structures begin to predominate.

Additionally, FTIR spectroscopy clearly indicates that NTBD molecules interact with both the hydrophobic and hydrophilic regions of the lipid membrane; however, interactions with the polar fragment appear to be significantly stronger.

## Data Availability

The original contributions presented in this study are included in the article/Appendix A. Further inquiries can be directed to the corresponding authors.

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
