# Peer review of "Advanced Spectroscopic Studies of the AIE-Enhanced ESIPT Effect in a Selected 1,3,4-Thiadiazole Derivative in Liposomal Systems with DPPC"

_ijms, 2025, doi:10.3390/ijms262110643_

Round 1
Reviewer 1 Report
Comments and Suggestions for Authors
This manuscript, titled "Advanced spectroscopic studies of the AIE-enhanced ESIPT effect in a selected 1,3,4-thiadiazole derivative in liposomal systems with DPPC," systematically investigates the photophysical behavior of the novel 1,3,4-thiadiazole derivative NTBD within DPPC liposomal systems, with a key focus on revealing the enhancement of the excited-state intramolecular proton transfer (ESIPT) effect by aggregation-induced emission (AIE). The research is rigorously designed, methodologically comprehensive, supported by substantial data, and presents clear conclusions, holding significant value for understanding the behavior of such compounds in drug delivery systems. Minor Revision is recommended.
- The manuscript indicates that the AIE effect enhances the ESIPT process but provides insufficient explanation of the synergistic mechanism at the molecular level. It is recommended to supplement the discussion with an in-depth analysis of how the aggregated environment reduces the energy barrier for the enol-keto tautomerism by restricting intramolecular rotation or vibration.
- Currently, the enhancement of ESIPT by AIE is primarily described qualitatively through spectral changes. It is suggested to supplement with quantitative indicators, such as fluorescence quantum yield or ESIPT efficiency under different aggregation states, to more visually demonstrate the enhancement effect.
-
The reference formatting is inconsistent in parts. It is recommended to standardize the reference format throughout the manuscript, ensuring consistent formatting for authors, journal names, volume/issue numbers, and page numbers. Furthermore, regarding AIE, it is suggested to cite relevant literature such as Chin. Chem. Lett., 2023, 34, 108452 and Chin. Chem. Lett., 2023, 34, 108088.
Author Response
Reviewer 1
This manuscript, titled "Advanced spectroscopic studies of the AIE-enhanced ESIPT effect in a selected 1,3,4-thiadiazole derivative in liposomal systems with DPPC," systematically investigates the photophysical behavior of the novel 1,3,4-thiadiazole derivative NTBD within DPPC liposomal systems, with a key focus on revealing the enhancement of the excited-state intramolecular proton transfer (ESIPT) effect by aggregation-induced emission (AIE). The research is rigorously designed, methodologically comprehensive, supported by substantial data, and presents clear conclusions, holding significant value for understanding the behavior of such compounds in drug delivery systems. Minor Revision is recommended.
Authors’ response:
We sincerely thank the Reviewer for the positive and highly encouraging evaluation of our manuscript. We are grateful for all the comments and for the recommendation to make minor revisions. All remarks and suggestions have been carefully analyzed and incorporated into the revised version of the manuscript. We believe that the modifications introduced in response to the Reviewer’s valuable feedback have further improved the clarity and scientific quality of the work. The changes in the manuscript have been made in “track changes” mode, and the newly added sections are clearly highlighted in yellow.
- The manuscript indicates that the AIE effect enhances the ESIPT process but provides insufficient explanation of the synergistic mechanism at the molecular level. It is recommended to supplement the discussion with an in-depth analysis of how the aggregated environment reduces the energy barrier for the enol-keto tautomerism by restricting intramolecular rotation or vibration.
Authors’ response:
We thank the Reviewer for this very accurate and insightful comment. We fully agree that the AIE-enhanced ESIPT process requires a deeper explanation of its underlying mechanism. In the revised version of the manuscript, we have expanded the discussion to provide more details on the synergistic mechanism between molecular aggregation and the ESIPT process at the molecular level.
Added fragment:
Summarizing the spectroscopic data obtained in the liposomal environment, it is evident that in the aggregated state, NTBD molecules become significantly more rigid due to the restriction of their internal motions (rotational and vibrational). This intramolecular restriction (RIR/RIV) effectively suppresses non-radiative relaxation pathways (e.g., the TICT state), thereby maintaining the molecule in a configuration favorable for intramolecular proton transfer [„Towards trans-dual deuterated cyclopropanes via photoredox synergistic deuteration with D2O”]. As a result, the energy barrier for the enol–keto tautomeric transition is lowered, which enhances the emission of the keto form (increased ESIPT contribution). This observation is consistent with the RIM theory and the experimentally observed increase in fluorescence quantum yield at higher NTBD concentrations within the studied system. It should therefore be emphasized that molecular aggregation reduces the potential energy barrier associated with the tautomeric transition, facilitating the occurrence and detection of the ESIPT process even in environments markedly different from nonpolar solvents, where this phenomenon is more commonly observed.
- Currently, the enhancement of ESIPT by AIE is primarily described qualitatively through spectral changes. It is suggested to supplement with quantitative indicators, such as fluorescence quantum yield or ESIPT efficiency under different aggregation states, to more visually demonstrate the enhancement effect.
Authors’ response:
We thank the Reviewer for this very valuable and well-founded comment. In response to the Reviewer’s suggestion, fluorescence quantum yields were determined for NTBD incorporated into DPPC liposomes at different molar fractions (5–20%) in order to quantitatively assess the enhancement of the ESIPT effect in aggregated states. The selected concentration range was chosen to maximize the visibility of the observed effect. The results are presented in Table 1. Solutions of the reference fluorophore — Coumarin 1 — were prepared at concentrations corresponding to those of NTBD used in the liposomal samples shown in Figure 1 of the manuscript, and their fluorescence spectra were subsequently recorded. The corresponding emission data for Coumarin 1 are provided in the Supporting Information (Figure S2).
Based on these measurements, the fluorescence quantum yields of NTBD incorporated into liposomes were calculated according to Equation (6) described in the Materials and Methods section.
Introducing these quantitative fluorescence parameters complements the previous qualitative spectral observations and clearly demonstrates the aggregation-induced enhancement of the ESIPT effect, in full agreement with the Reviewer’s valuable recommendation..
- The reference formatting is inconsistent in parts. It is recommended to standardize the reference format throughout the manuscript, ensuring consistent formatting for authors, journal names, volume/issue numbers, and page numbers. Furthermore, regarding AIE, it is suggested to cite relevant literature such as Chin. Chem. Lett., 2023, 34, 108452 and Chin. Chem. Lett., 2023, 34, 108088.
Authors’ response:
We thank the reviewer for pointing this out. We have carefully checked and standardized the reference formatting throughout the manuscript, ensuring consistent style for authors, journal names, volume/issue numbers, and page numbers. In addition, we have included the suggested references related to AIE (Chin. Chem. Lett., 2023, 34, 108452 and Chin. Chem. Lett., 2023, 34, 108088) in the revised manuscript to provide a more comprehensive background.

Reviewer 2 Report
Comments and Suggestions for Authors
Dear Author,
I believe that this study is well thought out and organized, that it reveals original aspects and valuable contributions in the scientific field. Until now, numerous studies have been conducted on liposomal formulations as drug delivery systems and as an increase in the bioavailability of encapsulated drugs, but for DPPT derivatives, the one addressed in this paper is new. However, I believe that some aspects need to be clarified.
- It is known that DPPC provides good stability and limits drug leakage especially at body temperature, where DPPC liposomes are in a rigid, solid phase. It would be interesting to prepare another liposomal formulation with the composition of the lipid membrane DPPC and another unsaturated lipid in order to see the difference in the release profile of the encapsulated substance, a very valuable information for controlled release and especially Thermosensitive release.
- The introduction is too lengthy, covering fundamental background information about the importance of this study, the characterization of liposomal formulations that include the synthetic substance 1,3,4-thiadiazole derivative. This redundancy in information can tire the reader, which is undesirable for such a valuable article. Please try to rewrite the introduction to focus on the main objective of the current study.
- To the DLS analysis, I consider that information could be added about the zeta potential of the empty liposomes but also with different amounts of encapsulated synthesized substance and also data about PDI from the DLS analysis which is crucial for determining the uniformity of a particle dispersion size.
Author Response
Reviewer 2
Dear Author,
I believe that this study is well thought out and organized, that it reveals original aspects and valuable contributions in the scientific field. Until now, numerous studies have been conducted on liposomal formulations as drug delivery systems and as an increase in the bioavailability of encapsulated drugs, but for DPPT derivatives, the one addressed in this paper is new. However, I believe that some aspects need to be clarified.
Authors’ response:
We sincerely thank the Reviewer for the positive and highly encouraging evaluation of our work. We greatly appreciate the recognition of the originality and scientific value of our study, particularly regarding the novel application of the NTBD derivative within DPPC liposomal systems. We have carefully reviewed all of the Reviewer’s comments and have addressed each of them in detail in the revised version of the manuscript. The implemented changes have contributed to improved clarity and overall quality of the presentation. We are grateful for the constructive and thoughtful feedback, which has significantly strengthened our work. All changes in the manuscript have been made in “track changes” mode, with newly added sections highlighted in yellow.
- It is known that DPPC provides good stability and limits drug leakage especially at body temperature, where DPPC liposomes are in a rigid, solid phase. It would be interesting to prepare another liposomal formulation with the composition of the lipid membrane DPPC and another unsaturated lipid in order to see the difference in the release profile of the encapsulated substance, a very valuable information for controlled release and especially Thermosensitive release.
Authors’ response:
We sincerely thank the Reviewer for the valuable suggestion regarding the use of mixed lipid formulations incorporating an unsaturated phospholipid to better assess the influence of membrane fluidity on controlled and thermosensitive release. We fully agree that such a comparison would provide deeper insight into the behavior of the encapsulated compound.
At the current stage of the study, we have conducted experiments for liposomes composed of DPPC and DMPC; therefore, we are not able to include additional formulations containing unsaturated lipids in this manuscript. The measurements performed in DMPC were not included in the paper because they do not provide significantly different information regarding the investigated effects (ESIPT or aggregation-related phenomena) compared to the results obtained for DPPC. Of course, the DMPC data can be made available to the Reviewer upon request (they are included below in this response).
However, we fully agree with the Reviewer that this represents an important research direction, and we plan to conduct such experiments in future studies to investigate how the presence of unsaturated lipid components influences the release profile and ESIPT-related photophysical properties of NTBD. We also believe that the topic raised by the Reviewer is sufficiently significant to warrant a separate publication, so that the findings are not overshadowed by other results; this will certainly be one of our next research steps. Once again, we sincerely thank the Reviewer for this insightful and valuable comment.
Partial results for the second lipid are provided below.
Fig. Spectra (from top to bottom) of absorption, fluorescence emission, fluorescence excitation, and synchronous RLS for NTBD in a DMPC lipid environment at 20 mol% NTBD in DMPC, analogous to the results presented in the manuscript for DPPC.
- The introduction is too lengthy, covering fundamental background information about the importance of this study, the characterization of liposomal formulations that include the synthetic substance 1,3,4-thiadiazole derivative. This redundancy in information can tire the reader, which is undesirable for such a valuable article. Please try to rewrite the introduction to focus on the main objective of the current study.
Authors’ response:
We sincerely thank the Reviewer for this valid and very valuable suggestion. At times, one may tend to include too much information in the introduction, while our goal is, of course, to ensure the clarity of the work. We have carefully revised the introduction to reduce redundancy and to focus more concisely on the main objective of the present study. Background information has been streamlined to emphasize the rationale for using liposomal formulations with the 1,3,4-thiadiazole derivative, avoiding excessive detail. The revised introduction now provides a clearer and more focused context for the study. We hope that the changes we have implemented will significantly improve the clarity and highlight the main objective of this work.
In the manuscript, the revisions have been made in “track changes” mode, and all newly added sections are highlighted in yellow.
- To the DLS analysis, I consider that information could be added about the zeta potential of the empty liposomes but also with different amounts of encapsulated synthesized substance and also data about PDI from the DLS analysis which is crucial for determining the uniformity of a particle dispersion size.
Authors’ response:
We sincerely thank the Reviewer for this valuable and constructive suggestion. We agree that including zeta potential data could provide additional insight into the surface properties of the studied liposomal systems. However, the DLS instrument used in our study (Zetasizer Nano ZEN1600, Malvern Instruments Ltd.) does not have the capability to measure zeta potential, although some other models of this series do offer this option. We sincerely apologize, but such measurements could not be performed in the present work; in future studies, we will ensure this aspect is carefully addressed.
It is worth noting, however, that the NTBD molecule is nonionic and generally electrically neutral under the experimental conditions, so its incorporation is not expected to significantly affect the liposome surface charge or zeta potential values.
To complement the DLS analysis, we have included the polydispersity index (PDI) values, which reflect the uniformity of the liposome size distribution. These results have been added to the Supplementary Materials (Table S1) and are also presented for all liposome formulations in Figure 7, panel B of the main text.
The following sentence has also been added to the main text (Section 2.1, after Figure 7):
“Moreover, the PDI ranged from 0.2 for 5 mol% NTBD up to 0.68 for 15 mol% NTBD (Fig. 7B and Table S1 in the Supplementary Materials), indicating moderate uniformity of the liposome dispersions and confirming that NTBD incorporation did not significantly affect the overall stability of the liposomal vesicles.”
Table S1. Polydispersity index (PDI) values obtained from DLS analysis.
|
Sample |
PDI |
|
DPPC |
0.593 |
|
DPPC + 1% NTBD |
0.496 |
|
DPPC + 3% NTBD |
0.557 |
|
DPPC + 5% NTBD |
0.202 |
|
DPPC + 10% NTBD |
0.502 |
|
DPPC + 15% NTBD |
0.681 |
|
DPPC + 20% NTBD |
0.596 |
